# Navigating the Intersection of Glycemic Control and Fertility: A Network Perspective

**DOI:** 10.3390/ijms25189967

**Published:** 2024-09-16

**Authors:** Carlo Di Carlo, Costanza Cimini, Ramses Belda-Perez, Luca Valbonetti, Nicola Bernabò, Barbara Barboni

**Affiliations:** 1Department of Biosciences and Technology for Food, Agriculture and Environment, University of Teramo, 64100 Teramo, Italy; cdicarlo2@unite.it (C.D.C.); ccimini@unite.it (C.C.); rbeldaperez@unite.it (R.B.-P.); lvalbonetti@unite.it (L.V.); bbarboni@unite.it (B.B.); 2Department of Physiology, International Excellence Campus for Higher Education and Research “Campus Mare Nostrum”, University of Murcia, 30100 Murcia, Spain; 3Institute of Biochemistry and Cell Biology (CNR-IBBC/EMMA/Infrafrontier/IMPC), National Research Council, Monterotondo Scalo, 00015 Rome, Italy

**Keywords:** reproduction, glycemic control, infertility, immunity, advanced glycation end products, biological network, system biology

## Abstract

The rising incidence of metabolic diseases is linked to elevated blood glucose levels, contributing to conditions such as diabetes and promoting the accumulation of advanced glycation end products (AGEs). AGEs, formed by non-enzymatic reactions between sugars and proteins, build up in tissues and are implicated in various diseases. This article explores the relationship between glycemic control and AGE accumulation, focusing on fertility implications. A computational model using network theory was developed, featuring a molecular database and a network with 145 nodes and 262 links, categorized as a Barabasi–Albert scale-free network. Three main subsets of nodes emerged, centered on glycemic control, fertility, and immunity, with AGEs playing a critical role. The transient receptor potential vanilloid 1 (TRPV1), a receptor expressed in several tissues including sperm, was identified as a key hub, suggesting that the modulation of TRPV1 in sperm by AGEs may influence fertility. Additionally, a novel link between glycemic control and immunity was found, indicating that immune cells may play a role in endocytosing specific AGEs. This discovery underscores the complex interplay between glycemic control and immune function, with significant implications for metabolic, immune health, and fertility.

## 1. Introduction

Altered glycemic control in humans is a global public health emergency. The global diabetes prevalence rate in 2019 was 9.3%, and it could rise to 10.2% by 2030 [1]. Furthermore, more than 400 million individuals are affected by type 2 diabetes (T2DM) worldwide and over 1 million deaths are attributed to this disease every year, and numbers are still increasing [2]. In addition, the prevalence of people with impaired fasting blood glucose and impaired glucose tolerance, having abnormal blood glucose levels but not meeting the diagnostic criteria for T2DM, was estimated to be 22% in 2016 [3]. Similar epidemiological studies showed a rise in the global prevalence of metabolic diseases from 2000 to 2019, including obesity, hypertension, and non-alcoholic fatty liver disease [4]. The tight control of glycemia is essential to the normal functioning of the human body. Glycemia is maintained within a range of 4–6 mM through a sophisticated system of various hormones and neuropeptides released by the brain, pancreas, liver, intestine, adipose tissue, and muscle [5].

Impaired glycemic control can be related to reduced fertility, as elevated blood glucose levels can lead to irregular menstrual cycles, anovulation, and poor quality of oocytes [6]. It has a detrimental impact on women’s chances of pregnancy due to a reduced possibility of embryo implantation, as well as on the quality of oocytes and embryos in IVF patients [7,8]. For instance, a predominant cause of female infertility is polycystic ovary syndrome (PCOS), and a confirmed link has been found between PCOS and poor glycemic control [9]. PCOS affects also women who are diagnosed with T2DM in addiction to irregular menstrual cycles, with higher risks of suffering from anovulation [10]. Insulin resistance in T2DM hampers the proper functioning of the ovaries, thereby disrupting the hypothalamic–pituitary–ovarian axis leading to imbalanced hormone secretion and reduced oocyte quality [11]. In men, lower testosterone levels induced by T2DM cause erectile dysfunction and decreased motility, quality, or concentration levels of spermatozoa [12].

Advance glycation end products (AGEs), heterogeneous compounds formed by non-enzymatic reactions between reducing sugars and proteins, lipids, or nucleic acids, have attracted attention for many years due to their diverse biological implications in many pathological conditions, in particular in diabetic complications [13,14].

The formation of AGEs is closely linked to the metabolic milieu and oxidative stress, processes that are exacerbated under chronic hyperglycemia, such as in diabetes mellitus [15]. They disrupt the cellular function by altering proteins [16] and triggering inflammation through the receptor for AGEs (RAGE) [17]. AGEs are linked to various diseases, including cardiovascular issues [18] and neurodegenerative disorders such as Alzheimer’s [19], where they contribute to neuronal damage and cell death.

The female reproductive system is central to human reproduction. Oocytes originate from the ovaries and become mature throughout a woman’s life, culminating in ovulation and possibly fertilization [10]. Actualizing why it is necessary to understand the biology of oocytes and their dynamics within uterine environment is critical to reproductive health and disorders.

AGEs are increasingly recognized as contributors to fertility decline through various mechanisms. They accumulate in reproductive tissues, altering the structure and function of proteins essential for reproductive processes, leading to impaired oocyte quality, disrupted sperm function, and altered endometrial receptivity, all critical factors for successful conception [20]. AGEs also exacerbate oxidative stress and inflammation within the reproductive system, leading to compromised gamete quality and embryo implantation [21]. Additionally, AGE–RAGE interaction triggers signaling pathways that disrupt the hormonal balance and impair the reproductive function [22]. These cumulative effects underscore the significant role of AGEs in reproductive dysfunction and infertility.

Recent studies have begun to elucidate a possible link between AGEs and the activation of the transient receptor potential vanilloid 1 (TRPV1) channel in sensory neurons [23]. TRPV1, a member of the transient receptor potential (TRP) family of ion channels, is best known for its role in nociception and thermal sensation [24]. In addition to its traditional function in pain perception, recent evidence suggests that TRPV1 may play a broader role in neuronal physiology and pathology, including neurodegeneration, synaptic dysfunction, and neuroinflammation [25,26], all central features of neurodegenerative diseases.

In addition, the AGE–RAGE pathways may intersect with TRPV1 signaling cascades, amplifying neuronal dysfunction and accelerating disease progression [23].

The multifaceted role of TRPV1 makes it fundamental in sperm function and reproduction in general [27,28]. As reported by Ramal-Sanchez et al. (2021) in humans and distinct species, TRPV1 has been found in mature spermatozoa, performing foundational roles in sperm capacitation and the acrosome reaction. It responds to physical, chemical, and thermal stimuli that enable spermatozoa to migrate, thereby determining their movement direction towards the egg and its motility. The interaction of TRPV1 with endocannabinoids and its response to temperature gradients highlight its complex role in reproductive physiology.

Given the diverse functions of TRPV1 in sensory neurons and other cell types, it is plausible that it may also mediate the effects of AGEs on sperm function. However, the specific effects of AGEs on TRPV1 in spermatozoa and their implications for fertility remain largely unexplored.

Despite the significant advancements in the understanding of AGE biology, several key questions in the fertility field persist. Elucidating the precise molecular mechanisms underlying this AGE-mediated pathogenesis remains a critical challenge, necessitating interdisciplinary approaches integrating biochemical, structural, and computational methodologies. For this reason, the aim of this study was to propose a comprehensive network elucidating the intricate interplay between AGEs, reproductive processes, and immune regulation. Through the integrated analysis of molecular pathways, the research aim was to provide new insights into the role of AGEs in modulating fertility, ultimately contributing to a deeper understanding of reproductive health and immune-mediated fertility.

## 2. Results

Generation of AFIRNET and Identification of Three Different Subsets of Nodes Related to Glycemic Control, Fertility, and Immunity

All molecular data available to date were aggregated and the *AFIRNET* was constructed (see Figure 1 and Appendix A). The most relevant topological parameters for the network are listed in Table 1 (Appendix A).

The analysis of the *AFIRNET* revealed that the network topology fell into the category of Barabasi–Albert scale-free network [29]. In fact, the two necessary conditions for such a network were fulfilled—the presence of hyper-connected nodes (hubs) with a lot of nodes poorly connected (y = 29.693x^−1.276^, shown graphically in Appendix A), and the absence of a correlation between the node degree and clustering coefficient (R^2^ = 0.0017, shown graphically in Appendix A).

The network was represented using the Prefuse Force-Directed Open CL Layout, which organizes the network depending on the algorithm implemented as part of the Prefuse toolkit provided by Jeff Heer and highlights the underlying topology of the graph [30]. As is clearly displayed in Figure 1, the network is formed by three connected subsets of nodes. The first and second are related to the immune response and glycemic control, indicated by the green and red circles, respectively. Conversely, the third subset, related to fertility, is delineated by the purple circle.

The *AFIRNET* analysis revealed interactions linking AGEs, fertility-related factors such as TRPV1 activity, and immunity. Among all AGE receptors, TRPV1, cytokines (e.g., TNF-α, IL-6), and transcription factors (e.g., NF-κB) appear to be relevant, highlighting their role as key players in the interconnected pathways regulating reproductive health.

The PCA analysis highlighted a close relationship between the formation of AGEs and the activation of the receptors (RAGE and TRPV1)

The PCA underlined the relevance of different nodes in our network, such as RAGE, TRPV1, Fru-AGE, Glu-AGE, HbA1c (glycated hemoglobin), argpyrimidine, and glyceraldehyde. Indeed, they are placed in the graph separately from other nodes, which are inside the ellipse (Figure 2).

## 3. Discussion

The intricate interplay between AGEs, fertility, and immunity is a complex but crucial aspect of reproductive biology. This study used a network analysis approach to unravel the multiple connections between these factors and shed light on their collective impact on reproductive health.

The proposed model focuses on the potential interference of AGEs with TRPV1 receptors during sperm capacitation, aiming to elucidate their influence on fertility.

Looking for the most connected nodes in the *AFIRNET*, only the TRPV1 node was identified as a hub. This finding is very consistent with the centrality that TRPV1 appears to have in many biological contexts [27]. As TRPV1 has the third highest bottleneck score, just after RAGE and CML (the most known and studied AGE), TRPV1 is suggested to be a controller of the flow of information.

The results of the *AFIRNET* analysis have important implications for understanding the reproductive dysfunction associated with metabolic disorders, oxidative stress, and immune dysregulation.

The subset related to glycemic control revolves around dietary glucose and the pathways that lead to the synthesis of AGEs. The complex signaling of blood glucose homeostasis is regulated by central and peripheral mechanisms [31]. After a meal, a postprandial increase in blood glucose occurs, activating insulin secretion and glucose uptake by insulin-sensitive tissues. Nutrient ingestion also stimulates endocrine cells lining the gastrointestinal (GI) tract to release the incretin hormones augmenting insulin secretion [32].

Spinal and vagal afferents play a role in sensing and processing information from the GI tract and pancreas. The central nervous system (CNS) can influence the peripheral glucose function through neuroendocrine systems such as the hypothalamic–pituitary–adrenal (HPA) axis and autonomic nervous system (ANS). In fact, stressful stimuli can trigger the release of glucocorticoids, following the activation of the HPA axis, which stimulates glucose production by the liver. [32]. Importantly, the HPA and HPG (hypothalamic–pituitary–gonadal) axes are tightly linked to balance reproduction, guiding the metabolic system in stressful conditions [33].

When the complex control of glycemia is altered, leading to hyperglycemia, there is an elevated production rate of AGEs at cellular level, which act both extracellularly and intracellularly by binding to their respective receptors [34]. The biological effects induced by AGEs include the promotion of inflammation, thrombogenesis, and apoptosis [34,35]. The circulation of these compounds induces binding to their natural receptors, which are divided into different families [36,37]. The receptors are known to induce different responses, all related to glycemic control. The overproduction of AGEs causes dyshomeostasis and their increased concentration induces pro-inflammatory responses [38].

Glucose is transported from capillaries to interstitial fluid through simple diffusion, with the rate of delivery depending on the blood flow. The interstitial glucose levels are influenced by the rate of glucose diffusion, uptake by cells, metabolism, insulin levels, blood vessel supply, blood flow, and capillary permeability [39]. During a phase of hyperglycemia, it is possible that the composition of the organ fluids changes due to the presence of greater glucose and AGE levels; importantly, the composition of the oviductal fluid might also change. In vivo, life begins in the oviduct [40], and the oviductal fluid has a complex composition, which derives, among other sources, from a transudate of blood plasma and the secretions of the oviductal epithelial cells that form the lumen of the oviducts [41]. This results in the presence of molecules with an oviductal epithelial origin, as well as others derived from synthesis in other parts of the body [41]. In addition, oviductal fluid has enormous importance in reproduction, as it possesses the properties that are useful in conferring the fertilizing ability to spermatozoa [42]. Indeed, many studies have demonstrated the importance of oviductal bicarbonate, defining it as one of the main activators of capacitation by promoting membrane lipids [43].

Based on these premises regarding the diversity of the elements that form the oviductal fluid and their fundamental utility in reproduction, our interest was focused on AGEs in this work, on the assumption that they may actually be present in this fluid and that the fluid’s altered composition may affect its functionality.

In the context of reproduction, AGEs have already emerged as potential contributors to infertility and adverse pregnancy outcomes [44], and this underscores their importance in reproductive dysfunction.

By incorporating AGEs into the *AFIRNET*, their role as key mediators was elucidated, linking metabolic disorders to reduced fertility.

In the *AFIRNET,* the second subset of nodes identified was related to fertility, and the most connected node was TRPV1. TRPV1 emerged as a hub, highlighting its importance as a potential link between environmental stimuli, sperm physiology, and reproductive outcomes. In this portion of the network, TRPV1 relates to its involvement in key moments of the spermatozoa life cycle, including capacitation and the acrosomal reaction. TRPV1 is a homotetrameric membrane ion channel, structurally similar to voltage-gated ion channels (VGICs). The transmembrane helices S5 and S6 define the ion pore, while S1–S4 surround the pore and are involved in gating [45]. TRPV1 is a non-selective cation channel that is activated by noxious heat and protons, as well as by endogenous ligands, including capsaicin, with resultant sensations of heat and pain but also involved in thermoregulation [26,46]. This channel also participates in inflammation, where activation in the sensory neurons results in the release of neuropeptides, thereby further amplifying the inflammatory response [47]. TRPV1 has also been shown to be directly dependent on the concentration of AGEs, as the increased binding of these compounds to their receptor RAGE allows a greater influx of ions (including Ca^2+^) through TRPV1, resulting in altered cytosolic signals [23]. There is a strong link between TRPV1 and AGE–RAGE, confirmed by their involvement in diabetic neuropathy. Indeed, AGEs are well known to play a key role in the pathogenesis of this disease through their binding to RAGE and the resulting inflammatory response [48]. TRPV1 has been extensively studied in the field of neuropathy and has also been shown to be dysfunctional as an ion channel in experimental models of diabetic neuropathy [23]. The role of TRPV1 in spermatozoa is certainly not to be underestimated, since it is distributed in a highly regulated manner along the spermatozoon structure and is involved in the maintenance of Ca^2+^ homeostasis, as well as the flux of this ion in fertility processes such as capacitation and the acrosomal reaction [49]. TRPV1 is modulated by many factors, both exogenous and endogenous, as well as chemical and physical, all of them capable of interfering with its activity related to fertility in spermatozoa [27,50,51]. The clustering analysis showed a subset with TRPV1 in its center. In this portion of the *AFIRNET*, TRPV1 relates to its involvement in key moments of the spermatozoa life cycle, including capacitation and the acrosomal reaction. All endogenous and exogenous factors that positively or negatively regulate its activity are present.

In this complex scenario, the immune system has a strong role, as it is critically involved in the regulation of reproductive processes, including both protective and regulatory functions within the female reproductive tract [52]. By integrating immune factors into the *AFIRNET*, the complex immunological landscape of the female reproductive tract and its impact on fertility were captured.

For this reason, a relevant third and last subset in the *AFIRNET* about the immune response was identified. In this portion of the network, the action of SR-A is underlined, as it works as a receptor for AGEs present on antigen-presenting cells. SR-A/AGE complex leads to an immune response that promotes CD4+ T-cells. This process and the expected subsequent release of inflammatory cytokines [53] shows a highly specific immune response to AGEs. In fact, AGEs have already been shown to increase the polarization of macrophages towards the pro-inflammatory M1 phenotype following RAGE activation and subsequent NF-κB release [54], suggesting their role in immunity.

Additionally, AGEs themselves are capable, upon binding to their receptors, of triggering pathways that lead to the production of pro-inflammatory cytokines [55]. As it is well known that there is a strong relationship between immunity, inflammation, and fertility [56], a dyshomeostasis of these processes can lead to infertility. During the travel of the sperm cells within the female genital tract in mammals, there is a physiological and normal interaction between the immune system and male gametes; this was demonstrated to lead to a reduction in spermatozoa in a swine model [57], while also happening in humans [58]. Nevertheless, deviations in this immune reaction can trigger the production of antisperm antibodies, resulting in subfertility or infertility, an issue estimated to affect between 2 and 3% of women [52]. In swine, once a major part of the female genital tract (vagina, cervix, and uterus) has been traversed and upon reaching the oviduct, it does not seem to be leukocytes attacking the spermatozoa [59]. The fact that the oviduct seems to be a safer place for sperm cells is supported by the fact that the union between sperm cells and oviductal epithelial cells in vitro in bovine species upregulates the expression of anti-inflammatory cytokines (TFGB1 and IL10) and downregulates pro-inflammatory cytokines (TNF and IL1B), a process that could protect the sperm cells from an immune reaction [60].

On the other hand, the study performed by Martínez et al. (2007) shows that the exposition of spermatozoa to pro-inflammatory cytokines at inflammatory concentrations, together with leukocytes such as IL-8 and TNF-α, enhances the lipid peroxidation [61], presumably negatively affecting the function of sperm [62]. In addition, the in vitro coincubation of spermatozoa with inflammatory cytokines impairs the progressive motility of sperm [63]. It is known that epithelial cells in different organs express AGE receptors [64,65], and AGEs can be accumulated in the organism, affecting the circulation of inflammatory markers [66]. Although no studies have demonstrated the expression of AGE receptors in oviductal epithelial cells and their role in reproduction, the fact that they are expressed in other epithelial cells raises concerns about the effects that altered AGE levels may have on sperm.

Furthermore, evidence shows that AGEs can directly impact reproductive processes, including sperm function and ovarian health, potentially through mechanisms involving oxidative stress pathways where TRPV1 is implicated [67].

Finally, the PCA graphically helps in understanding the strong correlation of AGEs and their receptors. The separate nodes can be divided into two different categories, glycation products and receptors, highlighting how the formation of AGEs and the activation of RAGE and TRPV1 receptors are closely linked. The PCA also illustrates the importance of HbA1c in the network, which is currently used to monitor long-term glycemic control. In detail, our analysis of HbA1c in the blood provides data on the average blood glucose level during the last two to three months, i.e., the expected half-life in red blood cells [68]. Our results provide an intuitive overview of both the main players in our system and the possible markers for blood glucose monitoring.

## 4. Materials and Methods

To better explain the methodology used, a workflow is reported in Figure 3, concisely indicating the details of the approach followed in the main steps and reporting also main the software programs, bioinformatic tools, and general systems used. All steps are explained in this section of the manuscript.

Data Collection, Database Creation, Network Construction, and Network Analysis

To construct the network, the data were retrieved from manuscripts identified through a PubMed (https://pubmed.ncbi.nlm.nih.gov) search [69,70,71] using the following search terms: “TRPV1/advanced glycation end products ” and/or “advanced glycation end products/fertility” and/or “advanced glycation end products interacting receptors in sperm” and/or “effect of lack of glycemic control on fertility”. Studies from the last ten years were first considered using the PubMed tool “results by year”, defining the period from 2014 to 2024. Secondly, cited papers appearing in the references of these studies were selected only when their topic were referred to the defined keywords. To organize the information collected from the literature, the data were merged into a database created using Microsoft© Excel 365 Version 2408 (Appendix A). High-standard selection criteria were followed to make sure the data in this database were of great quality and reliability. These criteria included strict methodological requirements, the accuracy of the data, and their significance to our research goals [70,71,72]. To ensure good data strength, each article was reviewed according to validated quality protocols [70,71,72]. Specifically, we used thirty-two papers (PMID: 23767955, 35455991, 24624331, 10531386, 9065778, 2251254, 30766472, 29474476, 29930087, 27434539, 26578953, 21565706, 33890707, 34573117, 20607325, 35995334, 24114996, 29987748, 24231387, 35646963, 12473645, 8144582, 11035013, 15289604, 9604012, 16022682, 28427390, 26853630, 24505139, 18636435, 33919147, 12323090) referring to AGE synthesis, the biochemistry of the binding to AGE-specific receptors, the role of TRPV1 in spermatozoa, and the interaction of AGE to TRPV1 hyperactivation. The correctness of each element, syntax, and actual interactions reported in the scientific literature were confirmed.

The database was divided into the following sheets [70,71,72]: “ages”, containing the nomenclature of all known and studied AGEs; “receptors”, containing all AGE-specific receptors divided by class and their nomenclature; “AGE–TRPV1”, reporting all interactions through which the database was built (this sheet did not include all free and available molecules normally involved in enzymatic processes or reactions (e.g., H_2_O for hydration, Pi for phosphorylation, etc.)); “analysis”, reporting the analysis of the network by evaluating the number of nodes, node degree, node degree distribution, closeness, and centrality to evaluate the main characteristics of the network; “reference”, reporting the references used to create the database.

Once the database was created with the subdivisions just described, the *AFIRNET* (AGE, fertility, and immune response network) was created using Cytoscape© 3.10.1 (Cytoscape Consortium, http://www.cytoscapeconsortium.org) (Appendix A). Cytoscape© was employed as the primary software for network representation due to its proven effectiveness in visualizing and analyzing complex networks [30]. It provides detailed information on various parameters, as reported in Table 2. These metrics are essential for determining the type of network under study and offering insights into its structural properties and overall organization. The analysis was carried out by considering the network as undirected, assessing the topological parameters listed and described in Table 2, using the specific plugin NetworkAnalyzer by Cytoscape©.

In addition, the betweenness centrality and closeness centrality were computed. The betweenness centrality *C_b_(n)* of a node *n* is computed as follows:$$\sum_{s\neq n\neq t}\left(\sigma_{\mathrm{st}}(n)/\sigma_{\mathrm{st}}\right),$$
where *s* and *t* are nodes in the network differing from *n,*
$\sigma_{\mathrm{st}}$ denotes the number of shortest paths from *s* to *t*, and $\sigma_{\mathrm{st}}$
*(n)* is the number of shortest paths from *s* to *t* that *n* lies on. The betweenness centrality is only calculated for networks that do not contain multiple edges. The betweenness value for each node *n* is normalized by dividing the number of node pairs excluding *n*: *(N −* 1*)(N −* 2*)/*2, where *N* is the total number of nodes in the connected component to which *n* belongs. Therefore, the betweenness centrality of each node is a number between 0 and 1. The closeness centrality *C_c(n)_* of a node *n* is defined as the inverse of the average shortest path length and is calculated as follows:$$C_{c}(n)=1/avg(L(n,m)),$$
where *L(n,m)* is the length of the shortest path between two nodes *n* and *m.* The closeness centrality of each node is a number between 0 and 1. NetworkAnalyzer calculates the centrality of the proximity of all nodes and the trace with respect to the number of neighbors. The closeness centrality is a measure of the speed with which information spreads from a given node to the other reachable nodes in the network.

Identification of Hubs

We identified the hyperconnected nodes in the *AFIRNET*, i.e., the hubs, as described previously [69,71,72], using the following equation:ND>μ+σ,
where *ND* is the node degree, μ is the average node degree, and σ is the standard deviation of the node degree.

Identification of Bottleneck Nodes

Bottleneck identification within the *AFIRNET* was performed using the Cytoscape© plugin CytoHubba. It implements the following bottleneck calculation algorithm. Let *Ts* be the shortest path tree with a root at node *s*. *BN(v) =*
∑s∈V
*ps(v)*, where *ps(v)* = 1 if more than *|V(Ts)|/*4 paths from node *s* to the other nodes in *Ts* meet at vertex *v*; otherwise, *ps(v)* = 0 [73]. This parameter is a measure of the centrality of the node, expressed as the number of shortest paths in which the node *s* is located. It is related to the importance of nodes in controlling the flow of information within direct networks [74].

Statistical Analysis

The principal component analysis (PCA) was carried out by using Past 4.13 (Oslo, Norway). The PCA was performed to enhance the analytical power by capturing multiple parameters simultaneously. As it is a statistical multivariate analysis, a PCA enables the synthesis of complex data into principal components, thereby reflecting the variability of network parameters in a comprehensive manner [75].

## 5. Conclusions

In conclusion, the hypothesis that AGEs can affect fertility was proposed, suggesting AGEs as interferents with the physiological function of TRPV1 receptors expressed in spermatozoa during the capacitation process. This study provides a comprehensive view of the complex interactions between AGEs, sperm TRPV1 activity, immunity, and fertility. Further clinical and experimental validation studies are needed to confirm the results of our network analysis and translate them into effective strategies for treating reproductive disorders and improving fertility outcomes.

## Figures and Tables

**Figure 1 ijms-25-09967-f001:**
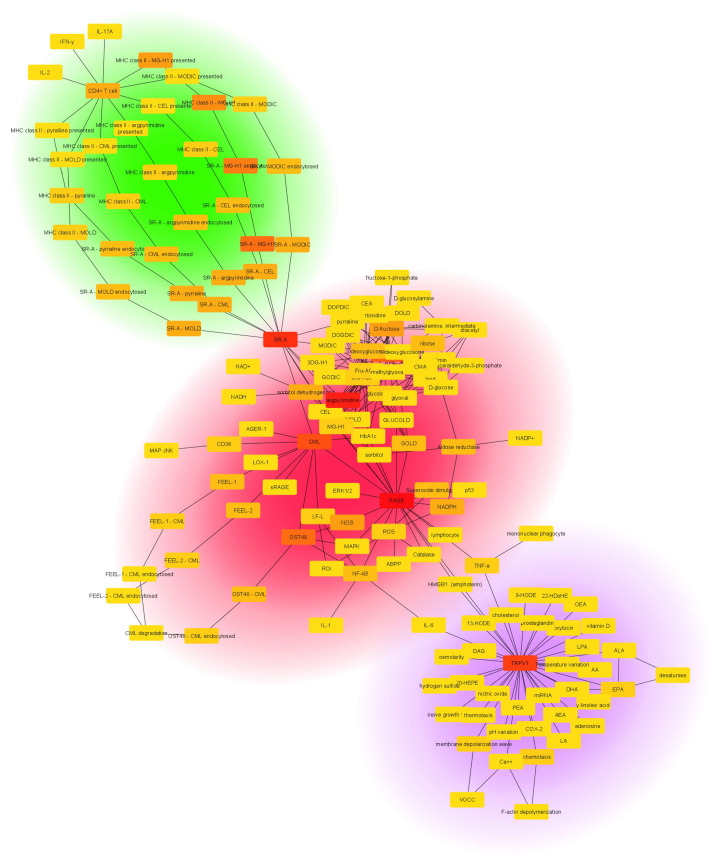
**AFIRNET (AGE, fertility, and immune response network).** The figure shows the *AFIRNET* and all of its data, created using Cytoscape©. By using the Prefuse Force-Directed Open CL Layout version 3.6.1, three different subsets of nodes can be identified, referring to glycemic control (circled in red), fertility (circled in purple), and immunity (circled in green).

**Figure 2 ijms-25-09967-f002:**
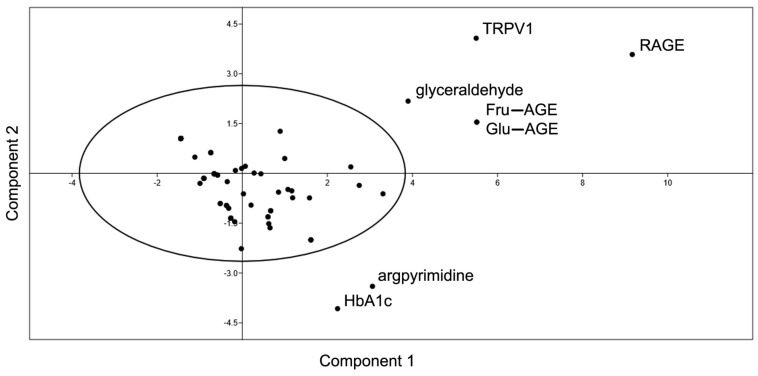
Principal component analysis. The PCA was performed by assessing the different parameters of the *AFIRNET* (average shortest, path length, betweenness centrality, closeness centrality, clustering coefficient, degree eccentricity, neighborhood connectivity, radiality, stress, topological coefficient).

**Figure 3 ijms-25-09967-f003:**
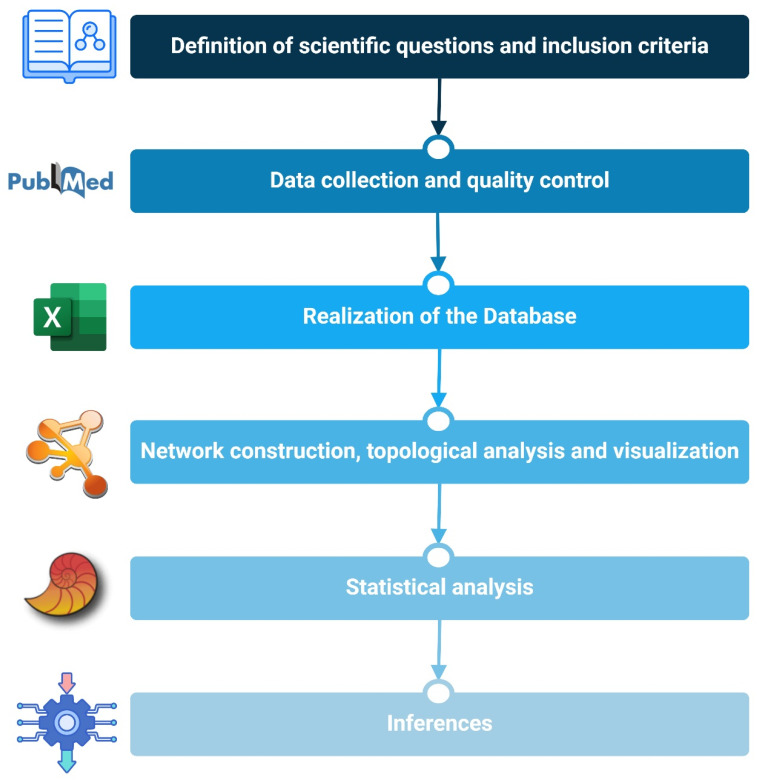
Methodology workflow, showing the bioinformatic steps of the study, indicating the software programs, tools, and general systems used.

**Table 1 ijms-25-09967-t001:** **Topological parameters of the *AFIRNET***. The table lists the most relevant topological parameters in **the***AFIRNET*.

Parameter		Value
Connected components		1
Number of nodes		145
Number of edges		262
Averaged number of neighbors		3.586
Clustering coefficient		0.023
Network diameter		16
Characteristic path length		5.453
Averaged number of neighbors		3.586
Node degree	ɣ r R^2^	−1.276
		0.8303
		0.6894

**Table 2 ijms-25-09967-t002:** Main topological parameters assessed in this study.

Parameter	Definition
Connected components	The number of networks in which any two vertices are connected to each other by links and which are connected to no additional vertices in the network
Number of nodes	The total number of molecules involved
Number of edges	The total number of interactions found
Clustering coefficient	Calculated as *CI =* 2*nI/kI(kI −* 1*),* where *n*I is the number of links connecting the *k*I neighbors of node I to each other. It is a measure of how the nodes tend to form clusters
Network diameter	The longest of all the calculated shortest paths in a network
Characteristic path length	The expected distance between two connected nodes
Average number of neighbors	The mean number of connections of each node
Node degree	The number of interactions of each node
Node degree distribution	Represent the probability that a selected node has *k* links
ɣ	Exponent of node degree equation
R	Pearson correlation coefficient of node degree vs. number of nodes on logarithmized data
*R* ^2^	Coefficient of determination of node degree vs. number of nodes on logarithmized data

## Data Availability

The original contributions presented in the study are included in the article/Appendix A, further inquiries can be directed to the corresponding author.

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
