# Peer review of "Navigating the Intersection of Glycemic Control and Fertility: A Network Perspective"

_ijms, 2024, doi:10.3390/ijms25189967_

Round 1

Reviewer 1 Report

Comments and Suggestions for Authors

The manuscript presents the results of a complex analysis of the bibliography on the relationship between poor glycemic control, protein glycation, and fertility. The introduction should reduce the concepts related to the effect of AGEs on other aspects of the body and go into more depth on those related to fertility, clarifying more explicitly the different ways in which these agents can affect fertility.

The materials and methodology are clear expose; it would only be important to indicate whether any exclusion criteria were used in the selection of the papers. The results are well presented, but it would be important to include a table summarizing the most important aspects for the reader unfamiliar with the methodology used. The discussion is clear and well founded.

Author Response

The Authors thank the Editor and Reviewers for their careful and rapid review of our manuscript. All the modifications suggested by the Reviewers have been appreciated and followed, and we consider that the manuscript has considerably improved now. Please find enclosed a clean version of the revised manuscript (IJMS_cleanversion) and a file with all the highlighted modifications (IJMS_highlighted). Lines of the manuscript in which the corrections are present are to refer to the file “IJMS_highlighted”. We really hope that you could reconsider our new version of the manuscript suitable for publication in IJMS.

We remain at your disposal in case you need any other clarification or additional information.

Looking forward to hearing from you, we send you our best regards.

Reviewer’s comments:

The manuscript presents the results of a complex analysis of the bibliography on the relationship between poor glycemic control, protein glycation, and fertility. The introduction should reduce the concepts related to the effect of AGEs on other aspects of the body and go into more depth on those related to fertility, clarifying more explicitly the different ways in which these agents can affect fertility.

Reply: Thank you for this important suggestion. We have revised the introduction to focus more specifically on the impact of AGEs on fertility, providing a clearer and more detailed explanation of the various mechanisms by which these agents influence reproductive health, while minimizing discussion of their effects on other bodily systems. Corrections in lines 74-89.

The materials and methodology are clear expose; it would only be important to indicate whether any exclusion criteria were used in the selection of the papers.

Reply: Thank you for your valuable feedback. We confirm that specific exclusion criteria were applied during the selection of the papers for this study to improve methodological rigor. Corrections in lines 140-146.

The results are well presented, but it would be important to include a table summarizing the most important aspects for the reader unfamiliar with the methodology used. The discussion is clear and well founded.

Reply: Thank you for your comment. We have added a summary workflow to highlight the key aspects of the methodology for clarity. Correction in lines 131-136.

Reviewer 2 Report

Comments and Suggestions for Authors

The manuscript conducts a thorough and detailed network analysis to explore the connections between AGEs, fertility, and the immune system. The primary shortcoming of the manuscript is that it almost exclusively focuses on data analysis. At our institute, several research groups are involved in network analysis and programming. Their reports typically provide more detailed explanations of the physiological properties and potential roles of the proteins found in the nodes than this manuscript. The current findings are more suited for hypothesis generation, and further research is needed to establish their practical applications.

I have a few minor suggestions and comments.

General Comments

11) The study aims to provide a comprehensive analysis of the relationship between advanced glycation end-products (AGEs) and fertility. It would be advisable to include a discussion on the female reproductive system, particularly the oocytes and ovaries, in the manuscript.

2)   2) A more detailed yet concise overview of the fertility of T2DM patients should be included.

Introduction

33) I suggest describing how impaired glycemic control is directly related to decreased fertility.

4  4) Elaborate on how AGEs and TRPV1 are linked to fertility.

Materials and Methods

55) What criteria were used to include data in the Excel database?

66) Provide a detailed rationale for choosing the specific methods (e.g., PCA) for network and statistical analysis.

Results and Discussion

77) TRPV1 plays a central role. I recommend at least a brief discussion of the physiological role of TRPV1 and how it may connect fertility and the immune system with AGEs.

It might be beneficial to separate the Results and Discussion into distinct sections. While this is not a requirement, I suggest it as a potential improvement.

Author Response

The Authors thank the Editor and Reviewers for their careful and rapid review of our manuscript. All the modifications suggested by the Reviewers have been appreciated and followed, and we consider that the manuscript has considerably improved now. Please find enclosed a clean version of the revised manuscript (IJMS_cleanversion) and a file with all the highlighted modifications (IJMS_highlighted). Lines of the manuscript in which the corrections are present are to refer to the file “IJMS_highlighted”. We really hope that you could reconsider our new version of the manuscript suitable for publication in IJMS.

We remain at your disposal in case you need any other clarification or additional information.

Looking forward to hearing from you, we send you our best regards.

Reviewer’s comments:

The manuscript conducts a thorough and detailed network analysis to explore the connections between AGEs, fertility, and the immune system. The primary shortcoming of the manuscript is that it almost exclusively focuses on data analysis. At our institute, several research groups are involved in network analysis and programming. Their reports typically provide more detailed explanations of the physiological properties and potential roles of the proteins found in the nodes than this manuscript. The current findings are more suited for hypothesis generation, and further research is needed to establish their practical applications.

I have a few minor suggestions and comments.

General Comments

  • The study aims to provide a comprehensive analysis of the relationship between advanced glycation end-products (AGEs) and fertility. It would be advisable to include a discussion on the female reproductive system, particularly the oocytes and ovaries, in the manuscript.

Reply: Thank you for your suggestion. We have revised the manuscript to include a discussion on the female reproductive system, focusing specifically on oocytes and ovaries, to provide a more comprehensive analysis. Correction in lines 74-89.

  • A more detailed yet concise overview of the fertility of T2DM patients should be included.

Reply: Thank you for the recommendation. We have added a detailed yet concise overview of fertility in T2DM patients to enhance the manuscript. Correction in lines 63-68.

Introduction

  • I suggest describing how impaired glycemic control is directly related to decreased fertility.

Reply: Thank you for your suggestion. We have included a description of how impaired glycemic control is directly related to decreased fertility in the revised manuscript. Correction in lines 58-60.

  • Elaborate on how AGEs and TRPV1 are linked to fertility.

Reply: Thank you for the recommendation. We have expanded the manuscript to provide a detailed explanation of how AGEs and TRPV1 are linked to fertility, exploring their roles and interactions in greater depth. Correction in lines 82-89 and 112-117.

Materials and Methods

  • What criteria were used to include data in the Excel database?

Reply: Thank you for raising this point. We have detailed the specific criteria used for data inclusion in the Excel database in the revised manuscript, outlining the selection process and relevant parameters. Correction in lines 148-151.

  • Provide a detailed rationale for choosing the specific methods (e.g., PCA) for network and statistical analysis.

Reply: Thank you for the request. We have included a detailed rationale for selecting specific methods, such as PCA, for network and statistical analysis in the updated manuscript, explaining their relevance and advantages for our study. Correction in lines 171-175 and 210-214.

Results and Discussion

  • TRPV1 plays a central role. I recommend at least a brief discussion of the physiological role of TRPV1 and how it may connect fertility and the immune system with AGEs.

Reply: Thank you for the suggestion. We have incorporated a detailed discussion on the physiological role of TRPV1, including how it may link fertility and the immune system with AGEs, to provide a more comprehensive understanding in the updated manuscript. Correction in lines 347 – 351 and 403-405.

It might be beneficial to separate the Results and Discussion into distinct sections. While this is not a requirement, I suggest it as a potential improvement.

Reply: Thank you for noticing. We have taken your advice and separated the Results and Discussion into distinct sections to improve the readability and structure of the manuscript. We believe this change will help clarify the presentation of our findings and their interpretation.

Round 2

Reviewer 2 Report

Comments and Suggestions for Authors

The requested changes were made with good quality.